# Advances in Targeted and Chemotherapeutic Strategies for Colorectal Cancer: Current Insights and Future Directions

**DOI:** 10.3390/biomedicines13030642

**Published:** 2025-03-05

**Authors:** Salique H. Shaham, Puneet Vij, Manish K. Tripathi

**Affiliations:** 1Medicine and Oncology ISU, School of Medicine, The University of Texas Rio Grande Valley, McAllen, TX 78504, USA; salique.shaham@utrgv.edu; 2South Texas Center of Excellence in Cancer Research, School of Medicine, The University of Texas Rio Grande Valley, McAllen, TX 78504, USA; 3Department of Pharmaceutical Sciences, St. John’s University, 8000 Utopia Parkway, Queens, New York, NY 11439, USA; puneet.vij08@outlook.com

**Keywords:** colorectal cancer, targeted therapies, chemotherapy, drug resistance

## Abstract

Colorectal cancer (CRC) remains one of the leading causes of cancer-related mortality worldwide, necessitating the continuous evolution of therapeutic approaches. Despite advancements in early detection and localized treatments, metastatic colorectal cancer (mCRC) poses significant challenges due to low survival rates and resistance to conventional therapies. This review highlights the current landscape of CRC treatment, focusing on chemotherapy and targeted therapies. Chemotherapeutic agents, including 5-fluorouracil, irinotecan, and oxaliplatin, have significantly improved survival but face limitations such as systemic toxicity and resistance. Targeted therapies, leveraging mechanisms like VEGF, EGFR, and Hedgehog pathway inhibition, offer promising alternatives, minimizing damage to healthy tissues while enhancing therapeutic precision. Furthermore, future directions in CRC treatment include exploring innovative targets such as Wnt/β-catenin, Notch, and TGF-β pathways, alongside IGF/IGF1R inhibition. These emerging strategies aim to address drug resistance and improve patient outcomes. This review emphasizes the importance of integrating molecular insights into drug development, advocating for a more personalized approach to combat CRC’s complexity and heterogeneity.

## 1. Introduction

Over several decades, cancer has remained one of the most formidable health challenges worldwide. According to the American Cancer Society, 9.7 million people died from cancer globally in 2022, while an estimated 20 million new cases were identified. It is anticipated that there will be 611,720 cancer-related fatalities and 2,001,140 new cancer cases in the US in 2024 [1]. There are several types of cancer, and colorectal cancer (CRC) is one of the deadliest. CRC is the second-most-common cause of cancer-related deaths globally and the third-most-common type of cancer overall [2]. American Cancer Society estimated 152,810 new cases and 53,010 deaths in 2024. Patients with CRC have a 5-year survival rate of 65% following surgery or surgery plus adjuvant chemotherapy and radiation therapy. The creation of novel therapeutic strategies for metastatic colorectal cancer (mCRC), however, is crucial because this survival rate drops to 15% [2,3,4], and more research is still needed to create efficient methods for medical intervention [4].

A previously treated localized CRC (nonmetastatic or stage I–III) may result in tumors being found at a remote location after treatment, or it may manifest de novo at stage IV, which is characterized as metastatic illness or cancer that has migrated outside the original colon mass. Metastasis most frequently occurs in the peritoneum, liver, lung, and lymph nodes [5,6]. Patients with non-metastasized CRC are typically treated with surgery [4]. Following surgical resection alone, the 5-year survival rates are 99% for stage I, 68–83% for stage II, and 45–65% for stage III disease [7]. Table 1 presents an overview of the CRC stages and corresponding survival rates after surgery. Despite having relatively low local failure rates, colon cancer frequently results in systemic recurrence after surgery, which is frequently the cause of mortality [8]. Despite mortality after surgery, our main aim is to increase tumor shrinkage and decrease tumor growth and the spread of the tumor. Therefore, a multimodal approach incorporating radiotherapy, chemotherapy, and targeted therapy is needed.

Although surgical resection remains the primary treatment for non-metastatic CRC, the high rate of systematic recurrence and the poor survival outcomes in metastatic cases highlight a critical gap in effective long-term disease management. Conventional chemotherapeutic regimens offer limited efficacy in advanced stages, and resistance to standard therapies further complicates treatment. Additionally, targeted therapies have emerged as promising alternatives, but patient-specific genetic variations, adverse effects, and treatment resistance constrain their clinical utility. Refining existing therapeutic strategies and developing novel approaches to improve the survival rates and quality of life of CRC patients is urgently needed. The research questions we want to discuss are the following: What are the current advancements in chemotherapeutic and targeted therapies for colorectal cancer? And how do they compare in terms of efficacy and patient outcomes? What mechanisms contribute to therapy resistance in colorectal cancer, and how can novel treatment strategies overcome these challenges? How can personalized medicine and biomarker-driven therapies improve treatment precision and outcomes in metastatic CRC? What are the emerging trends in combination therapies? And how do they enhance tumor regression and prevent recurrence? What roles do immunotherapeutic approaches play in CRC treatment? And how can they be integrated with existing therapies for optimal effectiveness? By addressing these research questions, this review aims to provide a comprehensive analysis of the current landscape of CRC treatment while identifying future directions for improved therapeutic interventions.

## 2. Chemotherapy

Most patients with mCRC cannot be cured; however, recent developments in medication and multidisciplinary care have significantly improved survival [9]. One of the first chemotherapy medications with anticancer properties was 5-fluorouracil 5-FU [10]. 5-FU is a chemotherapy medication that is frequently used to treat a variety of malignant tumors, such as malignancies of the breast, pancreas, skin, stomach, esophagus, head and neck. Since the 1990s, oral and intravenous 5-FU or other fluoropyrimidines (FPs) have been the cornerstone of systemic treatment for CRC [11]. Although chemotherapy based on 5-FU is the main treatment for CRC, it has drawbacks, including systemic toxicity, poor efficacy, selectivity, and resistance development. Capecitabine, a prodrug form of 5 FU, was created to address these issues, meet the demand for more convenient therapy, and enhance intratumor drug concentration levels, safety, and tolerability by converting tumor-specifically to the active 5 FU drug [12]. In 1996, irinotecan received approval in the United States for the treatment of mCRC that was not responding to 5-FU. It was then approved for use as a first-line treatment for mCRC in combination with 5-FU and leucovorin (LV) [13]. In patients with mCRC, oxaliplatin alone has shown moderate efficacy, with response rates (RR) ranging from 10% to 24% [14]. In contrast, the combination of oxaliplatin with 5FU has demonstrated RRs ranging from 20% to over 50% because of its synergistic effects [15]. The second approved oral medication is trifluridine plus tipiracil, which is intended for patients with mCRC who have already received fluoropyrimidine, chemotherapy based on oxaliplatin and irinotecan, an anti-VEGF biological therapy, and, if the patient has RAS wild-type, an anti-EGFR therapy [16]. Chemotherapy does have several drawbacks, though, including low tumor-specific selectivity, unpredictable innate and acquired resistance, unsatisfactory response rate, and systemic toxicity. So, there is a need to develop novel strategies to improve and replace current chemotherapy for tumor-specific target therapy for CRC. Table 2 summarizes the drugs used for CRC therapy and their modes of action, along with the year of approval.

## 3. Target Therapy

Targeted therapy is a type of cancer treatment that employs medications or other agents to specifically attack cancer cells, leading to their more precise destruction. This approach is designed to minimize damage to healthy cells in the patient. Targeted therapy can be effective through various approaches, including the following: (1) inhibiting or reducing the activity of cancer cells, (2) hindering or reducing the formation of a tumor’s blood supply, (3) aiding the immune system of a patient to identify and eliminate cancer cells, (4) identifying particular cancer cells and administering a medication directly to them [27].

### 3.1. Ongoing Target Therapy

#### 3.1.1. VEGF/VEGER Target Therapy

CRC is among the various cancers associated with angiogenesis. Angiogenesis refers to the development of new blood vessels, which originate from either endothelial progenitor cells or existing blood vessels. It plays a crucial role in various phases of cancer progression. Vascular endothelial growth factor (VEGF) plays a crucial role as an angiogenic factor in both primary and mCRC in humans [28]. This subgroup includes five proteins (VEGF-A, VEGF-B, VEGF-C, VEGF-D, and placental growth factor (PlGF) that play a crucial role in angiogenesis and lymph angiogenesis processes [29,30]. VEGF is detected early in the development of CRC, and its expression has been linked to a higher count of microvessels in colon tumors, which plays a role in the development of metastases. High expression of VEGF is also associated with recurrence after therapy. Studies show that anti-VEGF therapy leads to the regression of blood vessels, thus preventing tumor growth and the spread of cancer [31].

The VEGF signaling pathway in mammals comprises five glycoproteins that belong to the VEGF family. VEGF ligands interact with tyrosine kinase receptors, predominantly on vascular endothelial cells. VEGF-A, often known as VEGF, is the primary mediator of tumor angiogenesis. VEGF signals are primarily transmitted by VEGF receptor 2 (VEGFR-2). Subsequently, this leads to the activation of downstream signaling pathways such as the PLC-γ/PKC/Ras/Raf/MEK/MAPK signaling pathway [32,33]. This pathway is responsible for angiogenesis during CRC.

There are several anti-VEGF drugs which have been approved to cure CRC. Bevacizumab was the first VEGF-A-targeted humanized monoclonal antibody target drug developed as an antiangiogenic therapy [34,35]. Bevacizumab is a monoclonal antibody that plays a crucial role in targeting VEGF signaling pathways. It prevents the interaction between VEGF-A and its receptor, VEGFR, thereby inhibiting the activation of signaling pathways that promote neovascularization, which is the formation of new blood vessels. In vivo studies have shown that bevacizumab effectively inhibits the growth of blood vessels, induces regression of newly formed vessels, and normalizes the tumor vasculature. This normalization can enhance the delivery of cytotoxic chemotherapy to the tumor. Additionally, bevacizumab has been observed to exert direct effects on tumor cells, contributing to its therapeutic efficacy in the treatment of various cancers, including mCRC [36,37]. The combination of bevacizumab with irinotecan, 5-FU, and leucovorin (LV) has demonstrated an enhancement in the survival rates of individuals suffering from mCRC [38]. Aflibercept is a targeted therapy used in the treatment of CRC. It functions as a specific antagonist that binds to and inactivates circulating VEGF in both the bloodstream and the extravascular space. Structurally, aflibercept is a fusion protein that combines Domain 2 of VEGF receptor 1 (VEGFR1) and Domain 3 of VEGF receptor 2 (VEGFR2) with the Fc region of immunoglobulin G1 (IgG1). This design allows it to block all isoforms of VEGF-A, as well as PlGF. By inhibiting these factors, aflibercept plays a crucial role in preventing angiogenesis, which is vital for tumor growth and metastasis [39]. Regorafenib is a multi-kinase inhibitor that addresses several key hallmarks of CRC development by targeting various pathways critical for tumor growth and progression. Its mechanism includes anti-angiogenesis through the inhibition of vascular endothelial growth-factor receptors (VEGFR1, VEGFR2, and VEGFR3), TIE2, platelet-derived growth-factor receptor (PDGFR), and fibroblast growth-factor receptors (FGFR1 and FGFR2). This action helps disrupt the blood supply to tumors. In addition to its anti-angiogenic effects, regorafenib exhibits anti-proliferation properties by inhibiting kinases such as c-KIT, RAF1, BRAF, and RET, which are involved in cell signaling pathways that promote cancer cell growth. Furthermore, regorafenib plays a role in combating metastasis by inhibiting pathways related to cancer spread through the inhibition of VEGFR2, VEGFR3, and PDGFR. Finally, it also targets immune suppression within the tumor microenvironment by inhibiting colony-stimulating factor 1 receptor (CSF1R), which can enhance anti-tumor immunity. This broad range of inhibitory actions makes regorafenib a valuable therapeutic option for treating CRC, particularly in cases where conventional treatments may be less effective [40]. These anti-tumor properties are closely linked to the modulation of the tumor microenvironment (TME), which enhances therapeutic outcomes, even in cases of highly aggressive CRC. By altering the TME, these therapies can improve drug delivery, increase immune response, and reduce the overall tumor burden, leading to better patient prognosis. Targeting specific pathways involved in angiogenesis and immune suppression within the TME allows for a more effective strategy in managing CRC, especially in advanced stages [41,42]. Ramucirumab is distinct from other agents targeting the VEGF pathway, as it specifically binds to an epitope located on the extracellular domain of VEGFR-2. This binding effectively obstructs all VEGF ligands from attaching to this validated therapeutic target. As a result, ramucirumab prevents the binding of VEGF-C and VEGF-D to VEGFR-2, thereby inhibiting their activity and contributing to anti-angiogenic effects in the treatment of cancers, including CRC. This mechanism enhances its potential as a therapeutic option by targeting the angiogenic processes that support tumor progression and metastasis [27,43]. Table 3 lists the drugs targeting the VEGF signaling pathway.

Multiple factors contribute to resistance in VEGF-targeted therapy; these include Angiopoitin-2(ANG2). Patients with CRC who have elevated serum Ang2 levels showed a limited response to bevacizumab treatment, indicating that Ang2 is significant for the resistance mechanisms to anti-VEGF therapy [44]. Therefore, we require a dual approach that targets both Ang2 and VEGF signaling to effectively address CRC [45]. Numerous additional elements contribute to the resistance against VEGF signaling, such as FGF, IL-1, PDGF, PIGF, and TGF-β signaling. We require efficient, targeted therapy to combat resistance and tumors is required.

**Table 3 biomedicines-13-00642-t003:** Drugs used to treat against the VEGF signaling pathway, with mode of action and year of approval.

	Drug Targeted Against VEGF	Mode of Action	Response Rate	Year of Approval
1.	Bevacizumab	Bevacizumab is a humanized monoclonal antibody target against VEGF-A to prevent interaction with VEGFR-2.	~48.89% [46]	2004
2.	Aflibercept	Aflibercept inhibits VEGF-A,B and PIGF.	20.9% [47]	2011
3.	Regorafenib	Regorafenib inhibits VEGFR-1, -2, -3, PDGFR, c-Kit, and FGFR.	33% [48]	2012
4.	Ramucirumab	Ramucirumab is a human monoclonal antibody that targets the VEGFR-2 extracellular domain.	58.3% [27]	2014

#### 3.1.2. EGF/EGFR

The epidermal growth-factor receptor (EGFR) is a key member of the ErbB family of receptor tyrosine kinases (RTKs). These transmembrane proteins play crucial roles in cell signaling processes that regulate cell growth, survival, and differentiation. When activated, EGFR initiates a cascade of downstream signaling pathways that can promote cellular proliferation, angiogenesis, migration, survival, and adhesion. Activation occurs upon binding with specific peptide growth factors from the EGF family. Given the importance of these pathways for cancer cell survival, EGFR has emerged as a valuable target in the treatment of mCRC. By inhibiting EGFR activity, therapies can potentially disrupt these critical processes, leading to reduced tumor growth and improved patient outcomes [49,50].

The EGFR signaling cascade involves an adaptor protein complex that includes growth-factor receptor-bound protein 2 (Grb2) and son of sevenless (SOS). This complex plays a critical role in activating Ras-GTP by binding to phosphorylated tyrosine residues. Once Ras is activated, it triggers a cascading effect, activating RAF, MEK, and ERK through a series of phosphorylation events. The Ras–Raf–ERK signaling pathway is integral to regulating cell growth, differentiation, and survival. Dysregulation of this pathway can result in malignant transformations and tumor progression by promoting increased cell proliferation, prolonged survival, angiogenesis, resistance to apoptosis, invasion, and metastasis. As highlighted, the EGFR/MAPK signaling pathway is closely associated with oncogenic processes, making it significant in driving tumor growth and the progression of CRC. Research into targeting this pathway may provide valuable insights for developing therapeutic strategies against CRC [51,52]. Abnormal expression of this pathway has been identified as a target for CRC treatment [53,54].

There are multiple anti-EGFR medications which have been approved for the treatment of CRC. Cetuximab is a chimeric IgG1 monoclonal antibody that specifically targets the extracellular domain of the EGFR. By binding to EGFR, cetuximab inhibits ligand-induced receptor signaling, which helps modulate tumor cell growth. This mechanism of action is particularly significant in the context of certain cancers, including mCRC cancer, in which EGFR plays a critical role in tumor proliferation and survival. The use of cetuximab can enhance treatment efficacy, especially in patients with RAS wild-type tumors, providing a targeted therapeutic approach in combination with other treatments [55,56,57,58]. Combining cetuximab with immunotherapy and other targeted agents further expands the therapeutic landscape, offering renewed hope for patients with mCRC who encounter resistance to conventional therapies. This integrated approach enhances treatment efficacy and may lead to improved survival outcomes, particularly in those with RAS wild-type tumors. The synergy between cetuximab and other therapies can help overcome the limitations of single-agent treatments, providing a more comprehensive strategy to combat tumor growth and progression. As research continues to evolve, these combination strategies hold promise for better managing the complexities of mCRC [59]. Panitumumab is a human monoclonal antibody that specifically targets the EGFR [60]. By binding to EGFR, it effectively prevents the interaction of natural ligands, such as (EGF) and TGF-α, with the receptor. This blockade inhibits downstream signaling pathways that promote cell proliferation and survival, ultimately slowing down tumor growth and aiding in the management of certain types of CRC [61]. EGFR-targeting drugs are listed in Table 4.

Resistance arises in therapies targeting EGFR due to mutations in the MAPK signaling pathway. The emergence of acquired resistance to anti-EGFR treatments is still not well comprehended. Mutations in KRAS, NRAS, BRAF, MAP2K1, and the ectodomain of EGFR (EGFR-ECD) have been identified as contributors to acquired resistance [62]. We require a targeted therapy to address the EGFR pathway and to fight against resistance as well.

**Table 4 biomedicines-13-00642-t004:** Drugs used to treat against the EGFR signaling pathway, with mode of action and year of approval.

	Drug Targeted Against EGFR	Mode of Action	Response Rate	Year of Approval
1.	Cetuximab	Cetuximab is a chimeric IgG1 monoclonal antibody that specifically targets the extracellular domain of the epidermal growth-factor receptor (EGFR).	48.7–53.8% [63]	2004
2.	Panitumumab	Panitumumab is a human monoclonal antibody that specifically targets the EGFR.	10% [64]	2006

#### 3.1.3. HGF/CMet

The c-mesenchymal–epithelial transition (c-MET) is a receptor tyrosine kinase that plays a significant role in various cellular processes, including proliferation, survival, and motility [65,66,67]. It interacts with hepatocyte growth factor (HGF), which activates c-MET and drives signaling pathways that can lead to tumorigenesis [68].

The c-MET is a receptor tyrosine kinase that primarily binds to HGF [69]. These events trigger several downstream signaling pathways, such as the phosphoinositide 3-kinase/threonine-protein kinase (PI3K/AKT) pathway and the wingless-related integration site (Wnt) pathway, as well as others [70,71]. The activation of these pathways is crucial for mediating various cellular responses, including growth, survival, and migration, which can contribute to tumorigenesis. In the context of cancer, dysregulation of these signaling pathways is commonly observed and can lead to increased tumor progression, resistance to apoptosis, and enhanced metastatic potential [72]. Therefore, targeting these pathways may offer therapeutic strategies for managing cancers, especially those characterized by aberrant c-MET signaling and downstream effects [73]. The c-MET pathway is often implicated in the development and progression of several types of cancers, including CRC [73]. Cabozantinib is a small-molecule multiple tyrosine kinase inhibitor that disrupts signaling through various kinases, including VEGFR-2, MET, RET, KIT, AXL, TIE2, and FLT3 [74,75]. Since numerous tyrosine kinases frequently exhibit mutations and abnormal activation in tumors, such as those found in colon cancers, utilizing a small-molecule inhibitor to target them represents an ideal treatment approach for impeding tumor growth and survival, as well as the subsequent spread of cancer. Cabozantinib, a multi-tyrosine kinase inhibitor, has been observed to induce p53 upregulated modulator of apoptosis (PUMA), a crucial pro-apoptotic protein, through the activation of the AKT/GSK-3β/NF-κB signaling pathway in CRC. Upregulation of PUMA is crucial for the effectiveness of this chemotherapeutic drug. Crizotinib functions as an ATP-competitive small-molecule inhibitor targeting ALK, ROS1, and the Met/hepatocyte growth-factor receptor (HGFR). Crizotinib showed the ability to inhibit growth and enhance apoptosis in tumor cell lines containing ALK fusion variants (EML4-ALK or NPM-ALK) [76]. These medications demonstrate a promising targeted approach for treating CRC. Table 5 summarizes the drug targeting the HGF/cMet pathway.

#### 3.1.4. Hedgehog

The Hedgehog (Hh) signaling pathway maintains homeostasis in various tissues, including the musculoskeletal and digestive systems [80]. The Hedgehog (Hh) signaling pathway is a crucial extracellular morphogenic signal involved in various developmental processes. It plays multiple roles, including functioning as a mitogen to promote cell proliferation, acting as a cell survival factor to prevent apoptosis, and serving as an axon guidance factor to direct the growth of neurons during development [81,82,83,84]. The Notch receptors, designated as Notch1 through Notch4, are glycoproteins that span the membrane. Each Notch receptor consists of an extracellular portion, a transmembrane section, and a region that lies either within the membrane or in the cytoplasm and five ligands, Jagged-1, Jagged-2, Delta (DLL)-1, DLL-3, and DLL-4 [85]. When the Notch receptor interacts with one of its ligands, a series of proteolytic cleavages occur, initially facilitated by a metalloprotease and then followed by γ-secretase activity; this process leads to the release of an intracellular Notch (ICN) fragment that moves into the nucleus. Inside the nucleus, it binds with CBF-1 and MAML-1, forming part of a transcriptional complex [86]. The disruption of the Hh pathway has been associated with multiple cancers and developmental disorders, highlighting it as an important focus for potential therapeutic strategies. The Notch ligand Jagged1 is directly controlled by β-catenin, resulting in abnormal stimulation of Notch1 and 2 in CRC [87]. Notch-3 is observed to be elevated in mCRC and might influence the tumor development linked to CRC [88,89].

Aiming at the interval between the onset and the advancement of cancer and Hh signaling, targeted therapies have been created to block the pathway and its subsequent impacts. The premier Hh inhibitors available today include vismodegib and sonidegib. Vismodegib and sonidegib are targeted inhibitors of the Hh pathway that prevent Hh signaling by attaching to smoothened (Smo) and blocking the activation of downstream Hh target genes [90]. Hedgehog signaling pathway-targeting drugs are listed in Table 6. The schematic (Figure 1) shows the different pathways involved in targeted therapy for colorectal cancer.

### 3.2. Future Target Therapy

#### 3.2.1. IGF/IGF1R

The insulin and insulin-like growth factor (IGF) system is a complex signaling network that influences energy metabolism, cell growth, and the development of cancer [96,97]. The type 1 insulin-like growth-factor receptor (IGF-1R) is a glycoprotein that spans the membrane. It consists of two extracellular components and two cytoplasmic subunits that are receptor tyrosine kinases [98]. By interacting with IGF-1, IGF-2, and insulin, IGF-1R triggers downstream signaling pathways such as phosphoinositide 3 kinase (PI3K)/AKT and Ras/extracellular signal-regulated protein kinase (ERK) pathways [99], which are often activated in CRC [100]. In numerous cancer types, increased expression or activity of IGF-1R is frequently observed, and this is linked to the proliferation of tumor cells and their survival, resistance to apoptosis, and drug resistance [101]. IGF-1R has emerged as a focus for innovative treatments, particularly monoclonal antibodies and small molecules aimed at blocking tyrosine kinase activity [98]. As reported in a recent study, from 2003 to 2021, 16 IGF-1R inhibitors were included in 183 oncology clinical trials involving 12,396 patients [102]. Notably, none of these medications received approval for cancer therapy.

#### 3.2.2. Wnt/β-Catenin

The Wnt signaling pathway plays a role in various processes during embryonic development and the maintenance of tissue in adults, including cell growth, differentiation, homeostasis, and renewal, primarily by influencing gene transcription [103,104,105,106]. The Wnt signaling pathway has traditionally been classified into two primary types: the canonical pathway and the non-canonical pathway. The Wnt signaling pathway is critical in regulating various cellular processes, including development and tissue maintenance. The canonical pathway is primarily β-catenin-dependent, in which Wnt ligands activate receptors, leading to the stabilization and translocation of β-catenin to the nucleus, subsequently regulating gene expression involved in cell proliferation and differentiation. In contrast, the non-canonical pathway operates independently of β-catenin. This pathway is crucial for guiding cell movement and polarity during morphogenesis, influencing processes such as cell migration and organization without altering gene expression through β-catenin [107,108,109]. The Wnt/b-catenin pathway becomes active when a Wnt ligand attaches to the Frizzled (Fz or Fzd) receptor, which is a seven-pass transmembrane protein, along with its coreceptor. These occurrences result in the suppression of Axin-driven β-catenin phosphorylation, leading to the stabilization of β-catenin. Consequently, β-catenin accumulates and moves to the nucleus, where it forms complexes with TCF/LEF to activate the expression of Wnt target genes [107,109]. The overactivation and mutation of the Wnt/β-catenin signaling pathway has been linked to cancerous growth in the colon and rectum [108,110]. Approximately 1% of CRC cases exhibit activating mutations in the β-catenin protein [111,112], and elevated nuclear levels of β-catenin are linked to unfavorable outcomes in patients with CRC [113,114]. Nuclear β-catenin works in conjunction with TCF/LEF to trigger the expression of genes targeted by Wnt/β-catenin signaling, like c-MYC, cyclin D1, Matrix metalloproteinase-7, Musashi1 (Msi1), and EMT-related transcription factors [115,116,117,118,119,120]. Multiple phase I–II trials are in progress that investigate the use of a Wnt antagonist or modulator alongside chemotherapy agents (https://clinicaltrials.gov/, accessed on 15 December 2024). Research on CRC treatments focused on Wnt/β-catenin signaling currently includes natural substances, established medications, small molecules, and biological therapies [110]. Currently, no FDA-approved medications are specifically designed to block the Wnt/β-catenin signaling pathway for cancer therapy. Nonetheless, several FDA-approved drugs may be repurposed to target this pathway, like antiparasitic medications, such as artemisinin derivatives; quinine-related substances; benzimidazole derivatives; ivermectin; niclosamide; and so on [121]. Research is ongoing to explore the potential of these drugs in modulating Wnt signaling, which could provide new therapeutic strategies for treating cancers associated with dysregulation of the Wnt/β-catenin pathway. By focusing on existing medications, researchers aim to enhance treatment options for patients with Wnt-related cancers, including CRC. Several drugs currently undergoing phase I clinical trials aim to inhibit the Wnt/beta-catenin pathway, like OMP-18R5, OMP-54F28, LGK974, genistein, and resveratrol. There are several drugs in phase II, and the phase III trial is BB1608 [122].

#### 3.2.3. Notch

The Notch signaling pathway plays a critical role in regulating various cellular processes, including the self-renewal of stem cells, cell-fate determination of progenitor cells, and terminal differentiation of proliferating cells [123,124]. Four Notch genes have been recognized in mammals: Notch-1, Notch-2, Notch-3, and Notch-4. Five Notch ligands have been identified: Dll-1, Dll-3, Dll-4, Jagged-1, and Jagged-2. These molecules play a vital role in controlling cell fate decisions throughout development and in maintaining tissues [125]. Notch–ligand binding leads to the cleavage of the Notch receptor by metalloproteases and γ-secretase. This process is essential for the activation of the Notch signaling pathway. Upon ligand binding, the cleaved Notch intracellular domain (NICD) translocates to the nucleus; it establishes a ternary complex with a well-conserved transcription factor, CSL (CBF1/Suppressor of Hairless/Lag1), along with co-activators from the mastermind-like (MAML) family [126]. Multiple oncogenic pathways, including MAP Kinase, Akt, NF-κB, matrix metalloproteinases (MMPs), and mammalian target of rapamycin (mTOR) signaling, have been reported to engage in cross-talk with Notch signaling. This interaction suggests that these pathways collectively play a significant role in tumor aggressiveness in colon cancer [127,128,129,130]. In colon cancer, the expression levels of NOTCH1, NOTCH2, and NOTCH3 were found to be increased, whereas the expression of ATOH1 was decreased in CRC [131], Given that Notch signaling prevents the final differentiation of goblet cells within the colorectal mucosa, it plays an oncogenic role in CRC. Currently, there are no FDA-approved drugs specifically designed to target the Notch signaling pathway for the treatment of CRC. However, significant research efforts are underway to explore therapies that aim to modulate the dysregulation of the Notch signaling pathway to develop effective treatments for CRC [132]. The drug OMP-52M51, which targets Notch signaling, is currently undergoing a phase I clinical trial focused on mCRC, and several drugs targeting Notch are currently in the preclinical stage of development [133].

#### 3.2.4. TGF-β Smad

The transforming growth-factor-beta (TGF-β) signaling pathway is vital for regulating various biological processes, including tissue development and cell proliferation, differentiation, and apoptosis, as well as maintaining homeostasis within the body [134,135]. TGF-β signals are conveyed from the receptors on the cell membrane to the nucleus [136]. There are three types of TGF-β receptors in the TGF-β signaling pathway: TGFBR1, TGFBR2, and TGFBR3. Within the TGF-β signaling pathway, firstly TGF-β binds with TGFBR2 and forms a hetero-tetrameric complex between TGFBR2 and TGFBR1 [136]. TGFBR2 functions as an upstream kinase that phosphorylates TGFBR1 at the serine-rich GS motif, activating TGFBR1. Activated TGFBR1 triggers intracellular SMAD, and SMAD protein within the cell translocates to the nucleus from the cytoplasm and initiates transcription of the specific gene [137]. Alterations in TGF-β receptors and SMAD proteins are more commonly observed in CRC, leading to malignant characteristics, while mutations in TGF-β ligands are comparatively infrequent [136]. Currently, there are no FDA-approved drugs specifically designed to target the TGF-β signaling pathway for the treatment of CRC. However, some compounds and monoclonal antibodies are under trial to target the TGF-β signaling pathway to cure CRC [138].

In the past few decades, immune checkpoint inhibitors (ICIs) have emerged as a treatment option for tumors. Co-inhibitory receptors on T cell surfaces called cytotoxic T lymphocyte antigen 4 (CTLA-4) and programmed cell death 1 (PD-1) help to reduce T cell-mediated immune responses; however, cancer cells use these inhibitory molecules to encourage tumor tolerance and T cell exhaustion [139,140]. ICIs, consisting of Programmed Death 1 (PD-1), Programmed Death-Ligand 1 (PD-L1), or Cytotoxic T-Lymphocyte Antigen 4 (CTLA-4)-targeting monoclonal antibodies have significantly transformed the treatment strategies and outcomes for various solid tumors [141,142,143]. For the treatment of cancers, the US Food and Drug Administration (FDA) has approved PD-1 inhibitors (nivolumab, pembrolizumab, and cemiplimab), PDL-1 inhibitors (atezolizumab, durvalumab, and avelumab), and CTLA-4 inhibitors (ipilimumab) [144]. Antibody-drug conjugates are among the methods of treating cancer. (ADCs) represent an advanced category of targeted therapies that could enhance cancer treatment.

## 4. Discussion

CRC remains a critical global health challenge, with high mortality rates largely driven by the complexities of treating advanced and metastatic disease. Despite substantial progress in therapeutic strategies, current approaches—including chemotherapy and targeted therapies—continue to face significant hurdles, such as systemic toxicity, resistance, and limited tumor-specific efficacy. This discussion synthesizes current advancements and underscores the necessity of integrating molecular insights into future therapeutic strategies.

Chemotherapy has long served as the cornerstone of CRC treatment, with agents such as 5-FU, oxaliplatin, and irinotecan significantly improving survival rates. Nevertheless, the utility of these agents is constrained by the emergence of resistance and systemic adverse effects. Advances such as capecitabine, a prodrug of 5-FU, and synergistic combination regimens have mitigated some limitations but remain insufficient in achieving curative outcomes for mCRC. This underscores the urgency of enhancing drug delivery systems and developing novel strategies to overcome chemoresistance.

The advent of targeted therapies has marked a paradigm shift in CRC management, offering more precise interventions by focusing on molecular pathways central to tumor progression. Inhibitors targeting VEGF and EGFR pathways, including bevacizumab, cetuximab, and panitumumab, have shown substantial clinical success, particularly in RAS wild-type subsets. However, resistance mechanisms—arising from compensatory pathway activation or genetic alterations—have limited their long-term efficacy. This highlights the need for combination therapies and exploration of novel molecular targets.

Emerging pathways, such as Wnt/β-catenin, Notch, and TGF-β, hold considerable promise for CRC treatment. Additionally, The Werner Protein (WRN) is a multifunctional enzyme with both helicase and exonuclease activities. It has roles in various cellular processes crucial for maintaining genome stability, including DNA replication and transcription, DNA repair, and telomere maintenance. Several WRN inhibitors are being clinically investigated for treating patients with dMMR/MSI-H CRC [145,146,147]. Genomic and transcriptomic profiling plays a crucial role in shaping CRC treatment strategies. Liebs et al. [148] utilized droplet digital PCR (ddPCR) to detect common KRAS and BRAF point mutations in plasma; however, their findings indicated that detection was only sporadically successful in a cohort of stage I–III CRC patients. Other biomarkers, including MSI status, KRAS/BRAF mutations, and liquid biopsy approaches, are also discussed in detail [149]. These pathways play pivotal roles in tumor initiation, progression, and resistance, representing fertile ground for therapeutic innovation. Despite encouraging preclinical and early clinical evidence, no FDA-approved therapies targeting these pathways currently exist. The complexity of these molecular networks continues to pose challenges to clinical translation, necessitating further mechanistic research and clinical validation. Additionally, the tumor microenvironment (TME) plays a critical role in CRC progression and therapy resistance. Immune suppression within the TME, coupled with its influence on drug delivery and resistance mechanisms, presents an attractive target for therapeutic intervention. Agents such as regorafenib and cabozantinib, which modulate multiple aspects of the TME, have demonstrated promising results and warrant further clinical evaluation.

The future of CRC therapy lies in the adoption of precision medicine, leveraging advancements in genomic and transcriptomic profiling to tailor treatments to individual patients. The integration of predictive biomarkers for therapy selection, real-time monitoring of resistance, and rationally designed combination regimens targeting multiple pathways has the potential to revolutionize CRC management. Moreover, repurposing existing FDA-approved drugs for emerging molecular targets represents a cost-effective and efficient strategy for expanding therapeutic options.

In conclusion, significant strides have been made in improving CRC outcomes through advancements in chemotherapy, targeted therapy, and surgical interventions. However, critical challenges remain, particularly in overcoming therapy resistance, minimizing systematic and systemic toxicity, and enhancing long-term treatment efficacy. Addressing these challenges requires a deeper understanding of the molecular mechanisms underlying CRC progression, therapy resistance, and tumor microenvironment interactions. Future research must focus on the refining of current treatment strategies by answering certain key questions: What are the most effective targeted and chemotherapeutic approaches for CRC, and how do they compare patient outcomes? What mechanism drives the therapy resistance, and how can novel interventions counteract these effects? How can personalized medicine and biomarker-driven therapies be leveraged to improve treatment precision? What roles do combination therapies and immunotherapeutic approaches play in enhancing tumor regression and preventing recurrence? Achieving meaningful progress in CRC treatment will require a collaborative multidisciplinary effort, integrating molecular research clinical trials and personalized patient care. By addressing these fundamental research questions, we can drive innovation in treatment paradigms, ultimately improving survival rates and quality of life for individuals affected by colorectal cancer.

## Figures and Tables

**Figure 1 biomedicines-13-00642-f001:**
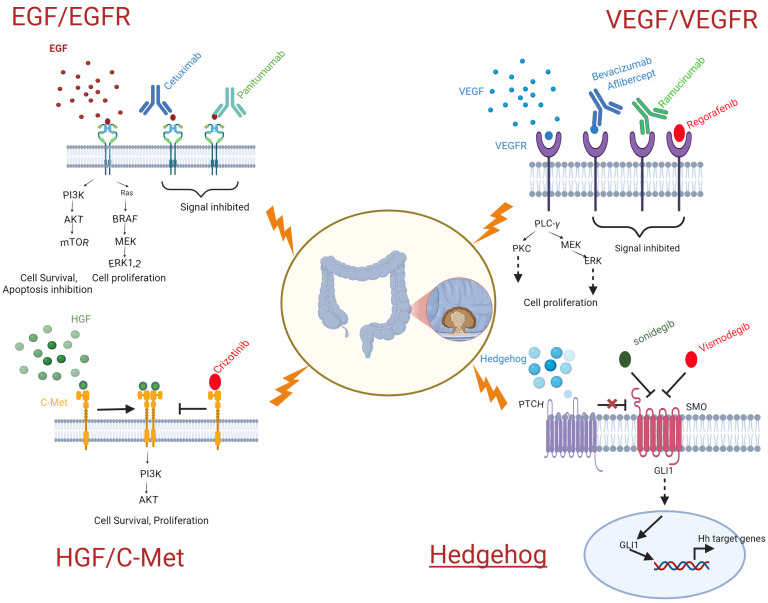
Target therapy for colorectal cancer.

**Table 1 biomedicines-13-00642-t001:** Colorectal cancer stage and survival rate after surgery.

Colorectal Cancer Stage	Survival Rate After Surgery
Stage I	95–99%
Stage II	68–83%
Stage III	45–65%

**Table 2 biomedicines-13-00642-t002:** Drugs used for colorectal cancer, with mode of action and year of approval.

	Drug Used for Colorectal Cancer	Mode of Action	Response Rate	Year of Approval
1.	5-Fluorouracil (5-FU)	5-FU exerts its anticancer effects by inhibiting thymidylate synthase (TS) and incorporating its metabolites into RNA and DNA [17].	54% [18]	1962
2.	Capecitabine (Xeloda)	Capecitabine was developed as a prodrug of FU, inhibiting thymidylate synthetase [19].	51–76% [20]	1998
3.	Irinotecan (Camptosar)	Irinotecan is a prodrug that inhibits DNA topoisomerase I [21].	19–32% [22]	1996
4.	Oxaliplatin (Eloxatin)	DNA replication and transcription disruption through intrastrand links between two adjacent guanine residues or a guanine and an adenine [23].	12–24% [24]	2002
5.	Trifluridine and tipiracil (Lonsurf)	TFD is a nucleoside analog; it is incorporated into replicating DNA strands, where it inhibits DNA synthesis and further cellular proliferation. TPI is an inhibitor of the enzyme thymidine phosphorylase, which is responsible for the breakdown of the active trifluridine component; thus, TPI boosts the levels of TFD [25].	40% [26]	2015

**Table 5 biomedicines-13-00642-t005:** Drugs used to treat against the HGF/CMet signaling pathway, with mode of action and year of approval.

	Drugs Targeted Against HGF/CMet	Mode of Action	Response Rate	Year of Approval
1.	Cabozantinib	Cabozantinib blocks c-Met, which in turn deactivates the SHH pathway, leading to a reduction in CRC tumor growth and angiogenesis [77]. Cabozantinib is a multi-tyrosine kinase inhibitor, utilized to induce p53 upregulated modulator of apoptosis (PUMA).	27.6% [78]	2021
2.	Crizotinib	Crizotinib targets ALK, ROS1, and the Met/hepatocyte growth-factor receptor (HGFR), and is used as an effective treatment against mCRC [79].		2011

**Table 6 biomedicines-13-00642-t006:** Drugs used to treat against the Hedgehog signaling pathway, with mode of action and year of approval.

	Drugs Targeted Against Hedgehog	Mode of Action	Response Rate	Year of Approval
1.	Vismodegib	Prevents Hh signaling by attaching to Smo and blocking the activation of downstream Hh target genes that inhibit the proliferation of tumors [91].	46% [92]	2012
2.	Sonidegib	Prevents Hh signaling by attaching to Smo and blocking the activation of downstream Hh target genes that inhibit cancer; also used as an alternative to vismodegib [93,94].	47.6% [95]	2015

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
