# Peer review of "Advances in Targeted and Chemotherapeutic Strategies for Colorectal Cancer: Current Insights and Future Directions"

_biomedicines, 2025, doi:10.3390/biomedicines13030642_

Round 1
Reviewer 1 Report
Comments and Suggestions for Authors
1-There are no clear research questions well-defined problem statements and gaps in the introduction. Can you clarify them and make sure to are as scope of your review.
2- the review included the existing studies on targeted therapy and chemotherapy for CRC. VEGF, EGFR, and Wnt signaling pathways were not clearly mentioned although it was discussed in many previous articles.
3- the methods section is not clearly outlined including eligible studies, and using settings, Can you please add them?
4-potential future therapeutic targets were discussed but the challenges in clinical treatments were not mentioned.
5-Strengthen your conclusion by answering the review questions based on previously mentioned studies.
6- there is an over-reliance on older references.
Author Response
Response to Reviewer 1 (Biomedicines-3480223)
We thank the reviewer for the positive response towards this review article, All 5 stars. We are answering the comments here:
- There are no clear research questions well-defined problem statements and gaps in the introduction. Can you clarify them and make sure to are as scope of your review.
Response: We thank the reviewer for the appropriate question. In the Introduction section, we have added a clear problem statement and research questions related to the current review. The changes are in BLUE.
- the review included the existing studies on targeted therapy and chemotherapy for CRC. VEGF, EGFR, and Wnt signaling pathways were not clearly mentioned although it was discussed in many previous articles.
Response: The Reviewer has raised a valid point, and we have added additional information in different sections. The changes are in BLUE.
- the methods section is not clearly outlined including eligible studies, and using settings, Can you please add them?
Response: Respectfully, we have submitted a review article, and since there are no experimental details, there is no method section.
- potential future therapeutic targets were discussed but the challenges in clinical treatments were not mentioned.
Response: We thank the reviewer for the critical suggestion, and in the current updated version, we have added the “Challenges” (as the development of resistance), where we have discussed the potential therapeutic targets. The changes are in BLUE.
- Strengthen your conclusion by answering the review questions based on previously mentioned studies.
Response: We are grateful for the suitable suggestion and have modified the conclusion to reflect the reviewer's comments. The changes are in BLUE.
- there is an over-reliance on older references.
Response: We have added new references, and the total number of references is 150; the latest been from 2025 (there were earlier 113 references). The changes are in BLUE.

Reviewer 2 Report
Comments and Suggestions for Authors
In this review, the authors assessed the current landscape of CRC treatment, focusing on chemotherapy and targeted therapies. Although chemotherapeutic agents, including 5-fluorouracil, irinotecan, and oxaliplatin, have significantly improved survival, challenges such as systemic toxicity and resistance are still unsolved. Targeted therapies, leveraging mechanisms like VEGF, EGFR, and Hedgehog pathway inhibition, offer promising alternatives, minimizing damage to healthy tissues while enhancing therapeutic precision. Although this review has some limitations, it is helpful in future directions in CRC treatment including development of innovative targets. Overall, this review paper is very interesting, but there are still several issues to be addressed before its acceptance.
Major comments:
- This review focuses on discussing the drugs which has high potential in the treatment of CRC. The drugs mentioned by the authors in table 5 and 6 don’t fit for this purpose. The authors should provide a clear explanation for this point.
- A few mono/bi-specific ADC drugs with high values for treating patients with mCRC are clinically being under investigation. The authors should add the related discussion in the manuscript.
- WRN is a multifunctional enzyme with both helicase and exonuclease activities and has roles in various cellular processes that are crucial for the maintenance of genome stability, including DNA replication, transcription, DNA repair and telomere maintenance. There are several WRN inhibitors being clinically investigated for treating patients with dMMR/MSI-H CRC. The authors should supplement the related discussion in this review.
Author Response
Response to Reviewer 2 (Biomedicine_3480223)
In this review, the authors assessed the current landscape of CRC treatment, focusing on chemotherapy and targeted therapies. Although chemotherapeutic agents, including 5-fluorouracil, irinotecan, and oxaliplatin, have significantly improved survival, challenges such as systemic toxicity and resistance are still unsolved. Targeted therapies, leveraging mechanisms like VEGF, EGFR, and Hedgehog pathway inhibition, offer promising alternatives, minimizing damage to healthy tissues while enhancing therapeutic precision. Although this review has some limitations, it is helpful in future directions in CRC treatment including development of innovative targets. Overall, this review paper is very interesting, but there are still several issues to be addressed before its acceptance.
Response: We sincerely appreciate your favorable remarks regarding the overall positivity of this review article and indicate that it will be helpful in shaping future treatment of CRC.
Major comments:
- This review focuses on discussing the drugs which has high potential in the treatment of CRC. The drugs mentioned by the authors in table 5 and 6 don’t fit for this purpose. The authors should provide a clear explanation for this point.
Response: The suggestion is very appropriate. We have now included the purpose of the drug in Tables 5 and 6. The changes are in BLUE.
- A few mono/bi-specific ADC drugs with high values for treating patients with mCRC are clinically being under investigation. The authors should add the related discussion in the manuscript.
Response: We thank the author for this specific suggestion. Immune checkpoint inhibitor therapy has now been included. The changes are in BLUE.
- WRN is a multifunctional enzyme with both helicase and exonuclease activities and has roles in various cellular processes that are crucial for the maintenance of genome stability, including DNA replication, transcription, DNA repair and telomere maintenance. There are several WRN inhibitors being clinically investigated for treating patients with dMMR/MSI-H CRC. The authors should supplement the related discussion in this review.
Response: We welcome the reviewer's suggestion. We have added the information PMID: 37517955, PMID: 38587317, and PMID: 39956276 to the discussion. The changes are in BLUE.

Reviewer 3 Report
Comments and Suggestions for Authors
Dear Authors,
Thank you for submitting your manuscript to Biomedicines. The topic of colorectal cancer treatment advancements is highly relevant and timely. The manuscript provides a comprehensive overview of current chemotherapeutic and targeted strategies, with a valuable discussion on emerging therapeutic directions. However, several areas require clarification, further development, and refinement to improve the manuscript's scientific rigor and readability.
- The manuscript contains some redundancies, especially in the Introduction and Discussion sections, where points about chemotherapy limitations and the promise of targeted therapies are repeated. Consider streamlining these sections for clarity and conciseness.
- While the manuscript outlines various signalling pathways (VEGF, EGFR, Wnt/β-catenin, etc.), there is a need for deeper mechanistic insights into how resistance develops within these pathways. For example, how do mutations in downstream effectors like KRAS affect the efficacy of EGFR inhibitors? Including more detail here will provide a richer narrative.
- Some sections lack references to the most recent literature (2023–2024). For instance, the section on Wnt/β-catenin could be strengthened by citing the latest clinical trials and emerging therapeutic agents currently in phase II/III trials.
- Tables 2–6 provide valuable summaries but lack uniform formatting and completeness. For instance, Table 3 lists VEGF-targeting drugs but omits critical efficacy data (e.g., progression-free survival or response rates from pivotal trials). Including such data would enhance the tables’ utility.
- Please consider adding a table summarizing key resistance mechanisms and corresponding strategies to overcome them, especially since resistance is a recurring theme in the manuscript.
- The Discussion section emphasizes the potential of future therapeutic targets like Notch, TGF-β, and IGF-1R. However, it would benefit from a more balanced view discussing the limitations and challenges in translating these preclinical insights into clinical practice.
- The concept of personalized medicine is introduced but not sufficiently developed. How can genomic and transcriptomic profiling concretely impact CRC treatment strategies? Providing specific examples or discussing biomarkers like MSI status, KRAS/BRAF mutations, or liquid biopsies would enhance this section.
- The manuscript contains several grammatical errors and awkward phrasings. For instance, in the Introduction: "Over several decades, cancer has been the most frightening disease in the world.” Consider rephrasing to: "Cancer has remained one of the most formidable global health challenges over the past decades." A thorough proofreading is recommended to improve overall readability.
I look forward to reviewing a revised version.
Author Response
Response to Reviewer 3 (Biomedicines-3480223)
Thank you for submitting your manuscript to Biomedicines. The topic of colorectal cancer treatment advancements is highly relevant and timely. The manuscript provides a comprehensive overview of current chemotherapeutic and targeted strategies, with a valuable discussion on emerging therapeutic directions. However, several areas require clarification, further development, and refinement to improve the manuscript's scientific rigor and readability.
Response: We thank the reviewer for their time and positive response to this review article. The suggestions are very helpful and have made this article much better.
- The manuscript contains some redundancies, especially in the Introduction and Discussion sections, where points about chemotherapy limitations and the promise of targeted therapies are repeated. Consider streamlining these sections for clarity and conciseness.
Response: We sincerely thank the reviewer for the positive comment; as per your suggestion, we have modified the Introduction and the discussion/conclusion section. The changes are in BLUE
- While the manuscript outlines various signalling pathways (VEGF, EGFR, Wnt/β-catenin, etc.), there is a need for deeper mechanistic insights into how resistance develops within these pathways. For example, how do mutations in downstream effectors like KRAS affect the efficacy of EGFR inhibitors? Including more detail here will provide a richer narrative.
Response: We thank the reviewer for the suggestion. We have included some resistance pathway information in the respective sections. The changes are in BLUE
- Some sections lack references to the most recent literature (2023–2024). For instance, the section on Wnt/β-catenin could be strengthened by citing the latest clinical trials and emerging therapeutic agents currently in phase II/III trials.
Response: We thank the reviewer for the observation. We have included the most recent references and drugs under trial. The changes are in BLUE
- Tables 2–6 provide valuable summaries but lack uniform formatting and completeness. For instance, Table 3 lists VEGF-targeting drugs but omits critical efficacy data (e.g., progression-free survival or response rates from pivotal trials). Including such data would enhance the tables’ utility.
Response: We thank the reviewer for the suggestion. We have included the efficacy data (response rate) in table 2-6. The changes are in BLUE
- Please consider adding a table summarizing key resistance mechanisms and corresponding strategies to overcome them, especially since resistance is a recurring theme in the manuscript.
Response: We thank the reviewer for the suggestion. The table might have been exhaustive. We have now included the resistance parameters at the appropriate section in the revised version. The changes are in BLUE
- The Discussion section emphasizes the potential of future therapeutic targets like Notch, TGF-β, and IGF-1R. However, it would benefit from a more balanced discussion of the limitations and challenges in translating these preclinical insights into clinical practice.
Response: We agree with the reviewer’s observation and have modified the discussion/conclusion section accordingly. The changes are in BLUE.
- The concept of personalized medicine is introduced but not sufficiently developed. How can genomic and transcriptomic profiling concretely impact CRC treatment strategies? Providing specific examples or discussing biomarkers like MSI status, KRAS/BRAF mutations, or liquid biopsies would enhance this section.
Response: We thank the reviewer for the suggestion. We have now Included it in the discussion; PMID: 31134762; PMID: 37569782. The changes are in BLUE
- The manuscript contains several grammatical errors and awkward phrasings. For instance, in the Introduction: "Over several decades, cancer has been the most frightening disease in the world.” Consider rephrasing to: "Cancer has remained one of the most formidable global health challenges over the past decades." A thorough proofreading is recommended to improve overall readability.
Response: Thanks for the insightful suggestion; we have checked the current submitted version with professional Grammarly and had a native English language Editor from the research office look at the m write-up. The changes are in BLUE

Round 2
Reviewer 2 Report
Comments and Suggestions for Authors
Following the authors' responses, the manuscript has been sufficiently improved to warrant publication in Biomedicines.